# Mouse Mammary Tumour Virus (MMTV) in Human Breast Cancer—The Value of Bradford Hill Criteria

**DOI:** 10.3390/v14040721

**Published:** 2022-03-30

**Authors:** James S. Lawson, Wendy K. Glenn

**Affiliations:** School of Biotechnology and Biomolecular Sciences, University of New South Wales, Sydney 2052, Australia; w.glenn@unsw.edu.au

**Keywords:** breast cancer, mouse mammary tumour virus, causation, transmission, oncogenic mechanisms, virus identification, case control studies, infection, multiple oncogenic viruses

## Abstract

For many decades, the betaretrovirus, mouse mammary tumour virus (MMTV), has been a causal suspect for human breast cancer. In recent years, substantial new evidence has been developed. Based on this evidence, we hypothesise that MMTV has a causal role. We have used an extended version of the classic A. Bradford Hill causal criteria to assess the evidence. 1. Identification of MMTV in human breast cancers: The MMTV 9.9 kb genome in breast cancer cells has been identified. The MMTV genome in human breast cancer is up to 98% identical to MMTV in mice. 2. Epidemiology: The prevalence of MMTV positive human breast cancer is about 35 to 40% of breast cancers in Western countries and 15 to 20% in China and Japan. 3. Strength of the association between MMTV and human breast cancer: Consistency—MMTV env gene sequences are consistently five-fold higher in human breast cancer as compared to benign and normal breast controls. 4. Temporality (timing) of the association: MMTV has been identified in benign and normal breast tissues up to 10 years before the development of MMTV positive breast cancer in the same patient. 5. Exposure: Exposure of humans to MMTV leads to development of MMTV positive human breast cancer. 6. Experimental evidence: MMTVs can infect human breast cells in culture; MMTV proteins are capable of malignantly transforming normal human breast epithelial cells; MMTV is a likely cause of biliary cirrhosis, which suggests a link between MMTV and the disease in humans. 7. Coherence—analogy: The life cycle and biology of MMTV in humans is almost the same as in experimental and feral mice. 8. MMTV Transmission: MMTV has been identified in human sputum and human milk. Cereals contaminated with mouse fecal material may transmit MMTV. These are potential means of transmission. 9. Biological plausibility: Retroviruses are the established cause of human cancers. Human T cell leukaemia virus type I (HTLV-1) causes adult T cell leukaemia, and human immunodeficiency virus infection (HIV) is associated with lymphoma and Kaposi sarcoma. 10. Oncogenic mechanisms: MMTV oncogenesis in humans probably differs from mice and may involve the enzyme APOBEC3B. Conclusion: In our view, the evidence is compelling that MMTV has a probable causal role in a subset of approximately 40% of human breast cancers.

## 1. Introduction

The underlying causes of breast cancer have been largely unknown. For many decades the betaretrovirus, mouse mammary tumour virus (MMTV), has been a causal suspect, but there has been limited evidence. However, in recent years, substantial new evidence has been discovered. Based on this evidence we hypothesize that MMTV has a causal role. We have used an extended version of the classic A. Bradford Hill causal criteria to assess this new evidence [1].

There have been a number of reviews of the potential role of MMTV in human breast cancer. Doubts have been expressed by Goedert [2], Park [3], Perzova [4], and Hameed [5]. The reasons for these doubts vary; however, the most frequent concern is the failure by many investigators to identify MMT in human breast cancers (these negative studies are listed in Appendix A in this current review). In complete contrast, Amarante et al. [6] conclude that there is convincing data in support of the association of MMTV with human breast cancer, although they indicate a need for further investigations of causal mechanisms. As outlined in our 2019 review, we also conclude that the data is convincing (Lawson and Glenn [7])).

The relevant evidence has been updated in this review. This includes six additional case control studies (Al Hamad [8], Periiera [9], Loutfy [10], Wang [11], Khalid [12], Gupta [13]), the identification of MMTV in the dental calculi of 4500-year-old women of the copper age (Lessi [14]), and the identification of MMTV gene sequences in the genome of women with breast cancer, but not in women with BRCA1 and 2 breast cancer [15].

*Risk factors.* There are well established risk factors for breast cancer. Being female is the most important. Additional risk factors include early age menarche and late age menopause. Early age pregnancy reduces the risk. There is an increased risk of breast cancer among women who migrate from low to high risk countries, for example, migration from Japan and China to the United States and Australia [16]. Inherited mutated BRCA1 and BRCA2 genes also increase the risk of breast cancer. None of these risk factors appear to be associated with MMTV.

*Terminology.* Although the term mouse mammary tumour virus is not scientifically correct (MMTV is not confined to mice, nor to breast and mammary glands, and has been present in humans for many thousands of years), we have chosen to use the term because it is familiar and has been used since 1966. With respect to this virus, several other terms have been used. These include mouse mammary carcinoma, MMTV-like virus, human betaretrovirus (HBRV), and human mammary tumour virus (HMTV). In the future, the scientific community may wish to consider the most appropriate terminology.

We have also chosen to use the term “breast” when referring to human breast cancer and “mammary” when referring to mouse mammary cancers.

*The A. Bradford Hill causal criteria.* These criteria have been immensely influential [1]. They have largely replaced the famous Koch postulates. The aim of the criteria is to discover whether an agent or pathogen is an association or the cause of a disease. In 1965, Hill developed nine criteria in the context of his research into the links between tobacco smoking and lung cancer. These were: 1. strength of association, 2. Consistency of association, 3. Specificity, 4. Temporality, 5. Biological gradient—dose response, 6. Plausibility, 7. Coherence, 8. Experiment, and 9. Analogy.

At that time, the role of viruses in various human cancers was not known. In addition, since 1965, there have been major developments in knowledge and technology. It has also been realised that the relevance of the individual criteria vary according to the nature of the pathogen or harmful agent. Accordingly, there has been a need to add and modify the classic Hill criteria. In response to the new evidence that viruses were an important cause of several cancers, Vladimir Vonka added a series of tests to the Hill criteria [17]. These additions included the ideas of Harald zur Hausen and others. The additions were: identification of the virus in cancer cells, epidemiology, oncogenicity of the virus in laboratory animals, and the capacity of the virus to transform normal cells into cancerous cells. We have added further criteria which include: means of transmission, oncogenic mechanisms, and the consideration of multiple causal pathogens.

It is arguable that Hill’s contributions to medical statistics and causal criteria form the basis of modern evidence-based science and medicine.

The list of these extended criteria, in order of importance with respect to MMTV and breast cancer, include:Identification and history of MMTV in human breast cancer.Epidemiology.Strength of the association between MMTV and breast cancer.Temporality (timing) of the association—evidence of infection by MMTV in normal tissues before the development of the cancer.Exposure. Does exposure to MMTV lead to infection, oncogenesis, and cancer?Experimental evidence—for example, the capacity of MMTV to cause cancer in experimental animals, the capacity to infect human cells, the ability of MMTV to transform normal human cells into malignant cells, and evidence that a vaccine or therapy can inhibit MMTV from infecting or transforming cells.Coherence, analogy—comparison of MMTV in human breast cancer with mouse mammary tumours.Transmission—identification of the source and means of transmission of MMTV to humans and specific tissues.Biological plausibility.Oncogenic mechanisms. While not necessarily helpful in determining causation, understanding oncogenic mechanisms is helpful for the development of prevention and treatment.Specificity. This criteria was in Hill’s original list but is rarely helpful as many viruses lead to cancer in many organs.

Hill strongly cautioned against dogmatism. ”None of my nine viewpoints can bring indisputable evidence for or against the cause-and-effect hypothesis and none can be required as a *sine qua non* (meaning an essential requirement). What they can do, with greater or less strength, is to help us to make up our minds on the fundamental question—is there any other way of explaining the set of facts before us, is there any other answer equally, or more, likely than cause and effect?” [1].

### 1.1. Assessment of the Role of MMTV in Human Breast Cancer

Identification of MMTV in human breast cancers.

In 1971, Moore D. et al., showed that some human milk contains particles physically identical to the mouse mammary tumour virus [18]. The historical electron microscopy-based images are shown in Figure 1. In 1972, Axel et al., identified the presence of particles in human breast cancer RNA homologous to particles of mouse mammary tumour virus RNA [19]. In 1987, Moore R. et al., used hybridisation techniques to identify the complete nucleotide sequence of the mouse mammary tumour virus [20]. This was an important development, as Crepin et al., Szakacs et al., and Wang et al., were later able to use these sequences to compare MMTV in mouse mammary tumours with MMTV in human breast cancers [21,22,23].

In 1980, Witkin et al., demonstrated that a significantly higher percentage of sera from breast cancer patients (26% of 54 breast cancer patients) contained IgG which reacted with MMTV, as compared to sera from women with benign breast disease (12% of 58 benign sera) or from normal controls (10% of 63 normal) [24]. Day et al., identified increased levels of the antibody to MMTV in 18.6% of US women with breast cancer, as compared to 2.8% of healthy women [25].

In 1995, the Pogo group from New York used polymerase chain reaction (PCR) technology to identify the MMTV *env* gene in 38.5% of 314 human breast cancer specimens, as compared to 7% of 29 benign human breast specimens, and in one of 27 normal human breast specimens [23]. Sequencing demonstrated 95 to 99% homology to the MMTV *env* gene. There was no homology with human endogenous retroviruses or other viral or human genes. Within a decade, these findings were confirmed by other research groups [26,27].

The complete nucleotide sequence of this virus was later amplified by the Pogo group from human breast cancers [28]. The 9.9 kb provirus was 95% homologous to MMTV, but only 57% homologous to human endogenous retrovirus [28]. This provirus displayed typical features of a replication competent virus. Melana et al., later showed that cultured human breast cancer cells produced viral particles which were similar to the mouse mammary tumour virus, with 85 to 95% homology with MMTV [29]. In addition, these particles had morphological features of betaretroviruses. In three independent studies, MMTV *env* gene sequences identified in human breast cancer were up to 97 to 100% homologous to MMTV *env* gene sequences [23,30,31].

MMTV replicates and spreads in human breast cells [32]. Faschinger et al., have demonstrated that MMTV is randomly integrated into the human genome in vitro [33]. Wang et al., identified integrations of MMTV in vivo in liver disease, proving that MMTV can definitely infect humans [34].

MMTV has been identified in human breast cancer in 41 different studies in 19 different countries. In 26 case control studies of breast cancer conducted by independent laboratories, MMTV has been identified in approximately 35% of breast cancer specimens, as compared to less than 2% in most control breast specimens (Table 1).

The Bevilacqua group of Pisa have identified MMTV sequences in dental material from 4500-year-old skeletons found in the Mediterranean [14]. The implication is that MMTV has infected humans since beyond human history.

Negative studies.

MMTV was not identified in 11 studies (Appendix A [3,4,50,51,52,53,54,55,56,57,58]). The viral load of MMTV in human breast cancers is very low. As a consequence, MMTV sequences are difficult to detect. Careful adherence to the most effective laboratory protocols is required. There are published critical appraisals of two of the negative studies. Stewart criticised the methods used by Zangen et al. [50,59] He showed that the temperatures used in the PCR procedures were too high. Pogo et al., criticised the methods used by Park et al. [3,60] Using the methods described by Park et al., Pogo could identify MMTV gene sequences but with much difficulty. In addition, Naccarato et al., have suggested that as the breast cancer specimens used in the Park et al., study were all from young women with a family history of breast cancer, negative MMTV findings would be expected [15]. This view is based on the recent study of sporadic breast cancer by Nacarrato et al., in which the prevalence of MMTV was 30%, as compared to familial BRCA breast cancer with a prevalence of 4% [15].

Challenges against the other negative studies have not been published. However, in the US, Australia, Italy, Mexico, and Iran, other studies have identified MMTV in breast cancers, despite the negative outcomes listed in Appendix A. Accordingly, the likely reason for the negative outcomes, at least in those countries, could be limited laboratory techniques.

Overall, the evidence is compelling that MMTV is present in over one-third of human breast cancers and is rarely present in normal breast tissues.

### 1.2. Contamination Issues

Due to the extreme sensitivity of PCR technology, contamination can be a problem. This contamination can occur from a wide range of sources, including contact by humans with MMTV positive rodents and their excreta, laboratory reagents, environmental contamination, and PCR carryover—the contaminants may be carried over from previous PCR procedures, contaminating pipettes, surfaces, and gloves.

However, it should be noted that a relatively high prevalence of MMTV DNA in control samples (that have different origins, such as tissue adjacent to the tumour, or breast tissue from reduction mammoplasty) does not constitute evidence of contamination [61]. Some healthy women harbor MMTV in breast tissue without having cancer. MMTV may be present in benign breast tissues, and 11% of healthy women have MMTV in their saliva [62]. We considered these issues in our study of MMTV in human breast cancer [38]. In that study, to exclude contamination, all reagents were shown to be free of MMTV-like sequences before use. In addition, PCR products were tested for the presence of mouse mitochondrial and genomic DNA. Similar procedures were followed by Mazzanti et al., in their study of MMTV in human saliva [62]. In neither study were mouse mitochondrial and genomic DNA identified.

MMTV sequences from the long terminal repeat (LTR) section of the MMTV genome have been identified in human breast cancers using next generation massive parallel sequencing (NGS) [63]. These MMTV sequences were highly homologous to the reference MMTV genome based on BLAST technology. Lehrer and Rheinstein have recently confirmed that MMTV gene sequences have not been identified in the human genome, but are present in breast tumours and normal breast tissues [64]. These data, based on techniques very different from PCR, confirm the identification of MMTV-like gene sequences in human breast tumours and add validity to PCR based studies.

In 2013, The Cancer Genome Atlas (TCGA) sequencing identified 3/800 MMTV positive breast cancers [65]. Zapatka et al., 2019 stated “a potential limitation of our analysis due to DNA and RNA extraction protocols are less likely to include DNA or RNA viruses.” They were only able to find one MMTV sequence in a renal carcinoma and none in 214 breast cancer samples [66]. It should also be noted that contamination frequently occurs using next generation sequencing techniques [67]. We are not aware if this problem has adversely affected any studies of MMTV and breast cancer.

NGS techniques are not as sensitive for the identification of retroviral nucleotide sequences as PCR [63,68]. This is the reason for the much more frequent identification of MMTV-like nucleotide sequences by PCR as compared to NGS.

If contamination had occurred in the 26 case control studies based on PCR, it would be expected that the positive identification would be similar in both cancers and controls (Table 1). As referred to above, MMTV gene sequences were identified, on average, in about 35% of breast cancer specimens, as compared to less than 2% in most control breast specimens.

Wang et al., found variability in LTR sequences that provides reassurance against viral contamination [34].

Using PCR, Naccarato et al., have demonstrated that MMTV *env* gene sequences can be identified in 30% of sporadic breast cancers, but only 4% of BRCA1 and 2 breast cancers [15]. This eliminates the possibility of murine contamination because if contamination had happened, the MMTV gene sequences would have been similar in both sets of breast cancers.

We conclude that the identification of MMTV in human breast cancers, benign samples, and normal breast tissues, is accurate in almost all studies.

## 2. Epidemiology

The incidence of breast cancer varies greatly between countries [69]. The Netherlands, France, the US, and Germany have high rates—between 85 and 95 cases per 100,000 age-adjusted females. Japan, China, and India have low rates—between 30 and 45 cases per 100,000 age-adjusted females. There are also marked differences in the incidence of breast cancer between Western and Eastern Europe. In the Netherlands, France, Italy, and the United Kingdom, the incidence of breast cancer is over 85, as compared to Russia and the Baltic countries, with an incidence of less than 50, per 100,000 age-adjusted females.

There are several reasons which may account for these differences in the prevalence of breast cancer. These include food consumption patterns that influence hormone levels. Hormones can influence the multiplication of oncogenic viruses such as MMTV. An additional reason is the distribution of MMTV-infected mice and other animals, including domestic cats and dogs. These influences are interrelated. They will be considered separately below.

There are strong associations between dietary patterns and levels of circulating estrogens, with energy rich diets correlated with high circulating estrogens [70]. In populations with a low risk of breast cancer, such as China, Japan, and Indonesia, the risk of breast cancer is up to seven times higher in women who consume the highest levels of fats and energy [71,72]. These findings parallel the consistent correlations between per capita fat and energy consumption and breast cancer risk between countries [73]. Endogenous estrogens are central to the aetiology of breast cancer; in the absence of estrogens, breast cancer does not occur.

Steroid hormones, including estrogens, influence MMTV oncogenesis in mice [74]. MMTVs contain hormone responsive elements, as does the MMTV virus in humans [28,74]. These hormone responsive elements are present in MMTVs amplified from human breast cancers [74]. There is a significant association between the prevalence of progesterone receptors and the MMTV-like virus in human breast cancers [75]. MMTV replicates and spreads in human breast cells under the influence of glucocorticoid hormones [32]. Pregnant women have high levels of a range of hormones. The prevalence of MMTV positive breast cancer in pregnant women is double that of non-pregnant women of the same ages [76].

Breast cancer incidence trends are also influenced by the distribution of mice and MMTV. Stewart et al., have demonstrated that, in Europe, these incidence trends significantly parallel the presence and distribution of the mice *Mus domesticus* and *Mus musculus* [77]. *Mus domesticus* are mainly located in Western Europe and *Mus musculus* in Eastern Europe. *Mus domesticus* appear to carry more infectious MMTV than do *Mus musculus*. This correlation between the prevalence of MMTV positive human breast cancer and MMTV infected mice of the *Mus domesticus* strain has been confirmed in China [11]. The prevalence of MMTV positive human breast cancer is significantly higher in northern China as compared to southern China. *Mus domesticus* mice are more prevalent in northern China. Wang et al., also showed there was a significant correlation between the prevalence of MMTV positive breast cancer in different countries and the global distribution of *Mus domesticus* [11].

Recent developments provide support for this hypothesis [55,77]. These include that mouse population outbreaks in Australia and New Zealand are correlated with spikes in breast cancer incidence [78].

### Migration and Breast Cancer

Historically breast cancer incidence has been over five times higher in the US, as compared to Japan and China. When Japanese and Chinese women migrate to the US, their breast cancer incidence rises over several generations and approaches that among US Caucasians [16,79]. The validity of these observations, now three decades old, has been recently challenged. Morey et al., have shown that Asian migrants to the San Francisco Bay area have a higher prevalence of breast cancer than US born Asian women [80]. In our view, the old studies remain valid because (i) they are consistent with studies of other populations and (ii) the participants in the old studies differ in socio-economic status from the new studies. Migrants from Japan and China to the US gradually increase their consumption of “festival” food from special to regular occasions, increase their consumption of fats and sugar, and decrease their consumption of carbohydrates and fiber [81,82]. Higher overall food consumption is correlated with increased sex and growth hormones [70]. As outlined above, the increased levels of these hormones are correlated with an increased prevalence of breast cancer, an increased activation of MMTV, and perhaps an increase in the activation of other oncogenic viruses as well.

## 3. Strength of the Association between MMTV and Human Breast Cancer—Consistency

This criteria can best be assessed by case control studies. The first case control study in which the identity and prevalence of MMTV in human breast cancers was compared to benign and normal breast specimens was conducted by Axel et al., in 1972 [19]. This study is remarkable for its techniques, foresight, and outcome. It is also remarkable for its obscurity and lack of influence on research into the causes of breast cancer. Using molecular hybridisation methods, MMTV involvement was determined to be positive in 19 (66%) of 29 breast cancers, with no involvement in 11 benign and 4 normal breast specimens [19]. These findings are broadly similar to those of the Pogo group more than two decades later [26]. The authors of the Axel study commented, “It is unnecessary at present to over-interpret the results presented. They clearly do not constitute definitive proof of a viral aetiology. They do, however, provide the most compelling evidence available for the involvement of virus-related information in human breast cancer” [19].

There have been 26 case control studies in which MMTV was identified (Table 1). MMTV was not identified in 11 additional studies. The case control studies compared the prevalence of MMTV in breast cancers with benign and normal breast specimens. With the exception of two early studies, all were based on PCR technology. Twenty two of these studies showed a consistent pattern of outcomes, namely, positive identification of MMTV in about 10 to 40% of breast cancers and zero to less than 5% in controls.

In Western women, the prevalence of MMTV in breast cancers is approximately 30 to 40%. In China and Korea, the prevalence of MMTV is approximately 10 to 20%. In 20 studies, the prevalence of MMTV in benign and normal breast tissues is consistently zero or below 5%. In four studies, the prevalence of MMTV in the control specimens was over 15%.

The implication of these outcomes is (i) there is a strong association between MMTV and human breast cancer and (ii) there is strong consistency in the findings in the studies conducted in 19 different countries.

### 3.1. Serology

The prevalence of MMTV antibodies in the serum of women with breast cancer is consistently five-fold higher than in the sera of women with benign breast conditions or in normal women (Table 2). Different methods have been used to measure the specificity of the antibodies in these studies. Regardless of the methods, the outcomes are similar, with the exception of Kovarik et al. [83] and Goedert et al. [2]. Kovarik et al., identified MMTV antibodies in 2 (3%) of 60 serum samples from women with breast cancer compared to 0 (0%) of normal controls. Goerdert et al., did not identify any MMTV antibodies in the sera of 92 women with breast cancer (however, MMTV gp52 was not identified in the positive controls). These negative findings have been superseded by Zhang et al., who used ELISA gp 52 to demonstrate MMTV antibodies in 10% of sera from women with breast cancer as compared to 2% in controls (*p* = 0.017) [84].

### 3.2. Genetics

Genetics plays a small but important role in breast cancer. Inherited mutations in the BRCA1 and 2 genes lead to a serious increased risk of breast cancer [96,97]. This accounts for approximately 3 to 5% of breast cancers. There is also an inherited susceptibility to develop breast cancer [27]. However, neither BRCA mutations nor inherited susceptibility directly cause breast cancer.

The prevalence of MMTV was compared in women with BRCA-associated breast cancer and women with sporadic breast cancer [15]. A total of 30% of 56 sporadic breast cancers contained MMTV sequences compared to 4% of 47 BRCA hereditary breast cancers (*p*< 0.001). These observations demonstrate that (i) MMTV is probably associated with human breast cancer, (ii) contamination by mouse material is excluded, and (iii) a role for endogenous betaretroviruses can be excluded.

## 4. Temporality (Timing) of the Association

MMTV has been identified in benign and normal Australian breast tissues up to 10 years before the development of MMTV positive breast cancer in the same patient [61]. This is a key causal criteria.

## 5. Exposure

Does exposure to the suspect virus lead to infection, oncogenesis, and cancer?

There are several lines of evidence which suggest exposure of humans to MMTV, and infection with MMTV, lead to the development of MMTV positive human breast cancer.

As outlined above, human breast cancer is more prevalent in locations where MMTV-carrying *Mus domesticus* feral mice are located [11,77]. This occurs in Western Europe and northern China [11,77].

A laboratory worker exposed to MMTV seroconverted, and antibodies to MMTV were found in the worker’s blood [98].

The saliva of newborns is MMTV-free, whereas MMTV is present in saliva of children (27%), healthy adults (11%), and breast cancer patients (57%) [62]. MMTV is also present in 8% of salivary glands [62].These observations suggest that exposure to MMTV via sputum leads to infection, and ultimately, to MMTV positive human breast cancer.

## 6. Experimental Evidence

(i). The capacity of MMTV to infect human breast epithelial cells.

MMTVs can infect human breast cells in culture [32,99]. Faschinger et al., demonstrated that when MMTVs infect human mammary epithelial cells, they randomly integrate their genomic information into the human genome of the infected cell [33]. Following integration of MMTV into the human genome of the infected mammary epithelial cells, the flanking sequences are of human and not mouse origin. This is an indication of an exogenous infection rather than contamination. The integration of MMTV into the human genome appears to be random and in multiple locations. Faschinger et al., also demonstrated that there is a similar pattern of random MMTV integration into the mouse genome [33].

(ii). Transformation.

Proteins expressed by the MMTV envelope gene are capable of malignantly transforming normal human breast epithelial cells [100].

(iii). MMTV is a likely cause of biliary cirrhosis [101]. Two randomised controlled trials using antiretroviral therapy (ART) have shown significant, but modest, improvements in hepatic biochemistry [102]. While these reports do not provide evidence that MMTV is causal in biliary cirrhosis, they do suggest a link between MMTV and disease in humans.

(iv). In experimental mice, tamoxifen therapy suppresses MMTV associated mouse mammary tumorigenesis [103].

## 7. Analogy

Many features of MMTV in mouse mammary tumours are almost identical to MMTV in human breast cancer. The life cycle of MMTV is similar in both mice and humans [104]. In humans, MMTV may be spread by sputum, human milk, or contaminated cereals. MMTV infects T and B lymphocytes in the human Peyer’s patches. MMTV is then activated by super antigens (SAgs). In humans, MMTV circulates in lymphocytes and then enters breast epithelial cells, where they integrate into the human genome [105,106]. In mice, lymphocytes may transmit MMTV throughout the mouse, but MMTV-initiated oncogenesis mainly occurs in the mammary glands [105].

Viral particles in human milk are physically similar to MMTV particles identified in mice [18]. MMTV gene sequences have been identified in human milk from normal lactating Australian and US women [107,108].

The nucleotide sequences and structure of MMTV-like viral sequences that have been identified in human breast cancer tissues are virtually identical to the MMTV sequences identified in mouse mammary tumours [29,109].

The same 63 cancer-related genes have been identified in both human breast cancers and mouse mammary tumours [110].

The MMTV envelope and capsid protein expression is similar in both mouse mammary tumours and human breast cancers [111,112].

MMTV super antigens (SAgs) play an essential role in MMTV-associated mouse mammary tumours [113] MMTV-infected B lymphocytes and dendritic cells lead to the expression of virus SAgs, which in turn stimulate T cells to produce cytokines that encourage the proliferation of infected B lymphocytes, thereby forming a reservoir of infected cells. MMTV is then conducted throughout the body by these lymphocytes, which facilitate the entry of MMTV into its target organ (the mammary glands or other organs). MMTV SAg is highly expressed in MMTV positive human breast cancer, and it appears to play a similar role in humans as it does in mice [114]. Human T cells respond to MMTV SAg [114].

MMTV infects intestinal T and B lymphocytes and randomly integrates into both the mouse and human genome [39]. MMTV-infected human lymphocytes can be found in women with MMTV positive breast cancer [21,115].

MMTV-associated tumour histology is similar in both mouse mammary tumours and human breast cancers [116]. It is of interest that this similarity mainly applies to mouse mammary tumours in mice who were infected with MMTV during the neonatal period (for more details, see the section on transmission below) [117]. However, the MMTV-associated histology is not necessarily specific to human breast cancer, as there are also similarities to other cancers, such as squamous cell skin cancers. A recent study has shed some light on this issue. Based on a study of genes and histology, Hollern et al., have shown that many features of murine tumour histology are conserved in both mice and humans among several different cancer types [118].

Antibodies to the MMTV surface envelope gp52 are present in the serum and mammary glands of feral mice [119]. Antibodies to MMTV gp52 have been identified in the serum of 2.8% of normal US women [25]. As indicated above, these findings have recently been confirmed by Zhang et al., who used ELISA to analyse gp 52 to demonstrate MMTV antibodies in 10% of sera from women with breast cancer, as compared to 2% in controls (*p* = 0.017) [84].

Mammary tumours in some strains of mice are hormone dependent [120]. The MMTV hormonal response elements in human breast cancers appear to promote cell growth, just as they do in mice [76].

In MMTV associated mouse mammary tumours, the oncogene Wnt-1 is highly expressed. In humans, the influence of MMTV on human breast cells leads to abnormally high Wnt-1 expression [41].

MMTV has been identified in mammary tissues in feral mice and in normal human breast tissues prior to the development of MMTV positive mouse mammary tumours and MMTV positive human breast cancers [61,121].

## 8. MMTV Transmission

The most plausible means of transmission of MMTV in humans is via human saliva [62]. MMTV gene sequences are present in saliva in 27% of normal children, 11% of normal adults, and 57% of women with breast cancer. MMTV-like gene sequences have been identified in human parotid glands—the source of saliva [62]. In mice, the main means of transmission of MMTV is via milk fed to newborn pups who develop mammary tumours as adults. The reason this means of transmission differs in humans is the destructive effect of human milk. This was shown experimentally by Sarkar et al. [122]. It has also been shown that MMTV can infect adult mice via nasal lymphoid tissue [123]. Although there is no direct evidence with respect to MMTV available, humans have well-developed lymphatic structures in the mouth and nose (tonsils and adenoids), which are possible entry points for MMTV.

As an example, the father, mother, and adult daughter of a family living together, all developed MMTV-associated breast cancer [124]. The MMTV env gene sequences identified in each family member was at least 98% homologous to the MMTV env sequences found in laboratory mouse strains. This observation supports the evidence that transmission of MMTV is likely to be by saliva. The prevalence of MMTV in human milk is significantly higher among women who are at greater than normal risk of breast cancer [108]. This suggests that human milk is a possible means of MMTV transmission.

MMTV gene sequences have been identified in mammary tumours in dogs and cats [125,126]. Saliva from dogs and cats is a likely mode of MMTV transmission. Women with companion dogs are at twice the expected risk of breast cancer [127].

The existence of US regulatory food standards that allow up to two pellets of rodent excreta per pint of wheat (US pint = 551 cm^3^) confirms the presence of mice in the modern human food chain [128].

## 9. Biological Plausibility

Retroviruses are the established cause of human cancers. Human T cell leukaemia virus type I (HTLV-1) causes adult T cell leukaemia, and human immunodeficiency virus infection (HIV) is associated with lymphoma and Kaposi sarcoma (however oncogenesis is largely due to its effect on the immune system). HIV appears to have crossed from chimpanzees and other primates to humans. Accordingly, it is plausible for MMTV to have crossed from rodents to humans (or from humans to rodents in past millennia), causing inflammation and cancer [129].

## 10. Oncogenic Mechanisms

The underlying causal mechanisms by which MMTV may cause human breast cancer are far from clear. With respect to laboratory mice, it has been demonstrated that the integration of MMTV proviral DNA into the mouse genome near one or more of the protooncogenes, such as Wnt-1 and Fgf, is associated with the development of mouse mammary tumours. However, more recent studies have indicated that MMTV oncogenesis in mouse mammary tumours is more complex than insertional oncogenesis. MMTV envelope genes encode the immunoreceptor tyrosine-based activation motif (ITAM) containing proteins, which are capable of malignantly transforming mouse mammary epithelial cells [100,112].

The relatively low levels of MMTV-like gene sequences detected in human breast cancers suggest that the virus is affecting oncogenesis by other mechanisms in addition to MMTV insertion. Additional mechanisms by which MMTVs can contribute to human breast oncogenesis include (i) proteins expressed by the MMTV envelope gene that are capable of malignantly transforming normal human breast epithelial cells [100], (ii) MMTV envelope protein p14 overexpression, which can function in an oncogenic capacity [130], (iii) MMTV encoded proteins (such as Rem, Sag, and Naf) or as yet uncharacterised proteins analogous to those of other complex retroviruses, such as Tax, may also have a role in breast cancer. The envelope proteins of Jaagsiekte sheep retrovirus (JSRV) which, like MMTV, is a beta retrovirus, can directly transform cells and offers a precedent for the ability of viral envelope proteins to malignantly transform cells [131]. There is also the intriguing possibility that MMTV and human endogenous retroviruses may interact, thus also playing a role [130].

Seven APOBEC3 genes, found on chromosome 22, are a group of cytidine deaminases that mutate cytidine to uridine [132]. All APOBEC3 genes target retroviruses by inserting mutations to inhibit replication, but this can also change cellular metabolism. Different APOBEC3 genes can have an opposing effect in breast cancer [133].

The enzyme APOBEC3B is an additional potential oncogenic mechanism. APOBEC3B is an enzyme which inhibits retrovirus replication. In mice, APOBEC3 has been shown to inhibit MMTV infections and viral replication, and to inhibit milk-borne MMTV virions [134]. In humans, inactivating mutations and deletions in APOBEC3B appear to play a role in breast cancer development [104,135]. These deletions, together with an increase in APOBEC 3A expression, increase the risk for breast cancer [136]. However, it should be noted that Gansmo et al., reported no association with breast cancer [137].

Human papilloma viruses have been shown to alter the expression of APOBEC3B, which may reduce its protective effects against MMTV [135,137,138,139]. It has also been shown that there is a significant increase in APOBEC-mediated mutagenesis in HER+/HER2 metastatic breast tumours, as compared to early stage primary breast cancer. Understanding the role of the APOBEC enzymes in breast cancer has become increasingly complex. Several APOBECs can promote cancer through mutagenesis, while at the same time, they can oppose cancer through an immune response [133].

Taken together, these studies suggest that the APOBEC family, and in particular APOBEC3A and 3B, may play a role in the early stages of breast cancer induction. Experimental data is required to confirm this.

## 11. Specificity

With regard to specificity, Hill’s statement “We must not, however, over-emphasise the importance of the characteristic” was prescient, as it is now well known that many viruses, for example human papilloma virus, are associated with several different cancers [1].

MMTV has been identified in many different cancers and liver diseases, including ovarian, prostate, endometrial, and skin, but not lung cancer [101,140].

## 12. Discussion

The Hill criteria has provided a means of systematically reviewing the evidence. With two possible exceptions, the evidence meets all of the extended Hill criteria. The first exception is that the oncogenic mechanisms for MMTV in human breast cancer are not clear (we have added this to the original criteria). An understanding of such mechanisms is not important in the search for causation, but it is relevant for the development of prevention and treatment. The second exception is that additional experimentation would be helpful. In 1948, Graff et al., injected virus like particles obtained from mice with mammary tumours into experimental mice who later also developed mammary tumours [141]. This proved that these viral particles, later identified as MMTV, were causal. It should be possible to conduct similar experiments with respect to human breast cancer.

### 12.1. Alternative Opinions

Several authors have argued against the involvement of MMTV in human breast cancer [2,3,4,5,142] (Goedert et al., 2006, Park et al., 2011, Perzova et al., 2017, Hameed et al., 2020, Gannon et al., 2018).

Goedert et al., did not identify serological activity against four strains of MMTV in US women with breast cancer [2]. This is the only negative outcome of 16 similar studies, as shown in Table 2. By way of contrast, MMTV antibodies were identified in the serum of 10% of 98 Canadian women with breast cancer, compared to 2% of controls, in a recent study by Zhang et al. [84]. This confirms the results of the earlier studies.

Park et al., did not identify MMTV *env* gene sequences in a series of Australian breast cancer specimens, whereas others had identified MMTV in 42% of Australian breast cancers [3]. Pogo et al., argued and demonstrated that Park et al., had used incorrect methods [60].

Perzova et al., argue that some MMTV positive human breast cancer results may be due to contamination with mouse DNA [4]. While contamination is always possible when highly sensitive PCR techniques are used, both Mazzanti et al. [62] and Nartey et al. [61] excluded mouse DNA contamination by performing murine mitochondrial DNA and intracisternal A-particle long terminal repeats PCR. All tested samples were free of mouse DNA.

Gannon et al., argue that a viral load <10 parts per million is not sufficient or capable of contributing to cancer development [142,143]. Vinner and colleagues have considered the reasons for the different outcomes of whole genome DNA sequencing as compared to DNA PCR [68]. The most important reasons are (i) the extremely small number of reads aligning to MMTV using next generation sequencing (9 reads out of more than 1.5 billion, based on Tang et al. [65], (ii) the MMTV genome is less than 10 kb and is only a minor fraction of the genome of the infected host cell, (iii) the infected cell type may constitute only a small fraction of the cancer specimen, and (iv) the infected cells may contain a low number of viral copies [68]. In addition, it cannot be assumed that mouse mammary tumour virus exerts oncogenicity by the continuous expression of viral gene products. For example, human papilloma viruses appear to exert an indirect influence on breast cancer by inhibiting the virus protective functions of the APOBEC3B enzymes [138].

There appear to be two main reasons for the differences in the published results concerning MMTV and human breast cancer. One, there are geographical variations in the incidence of breast cancer and the prevalence of MMTV, and two, because of the extremely low MMTV viral load, there are technical difficulties.

### 12.2. Multiple Viral and Causal Factors

A handful of viruses may collaborate to lead to oncogenesis in a single cancer. Hormones may play a role, with breast cancer as a classic example.

A. Bradford Hill stated, “I believe that multi-causation is generally more likely than single causation, though possibly if we knew all the answers we might get back to a single factor”, Hill 1965 [1].

There is substantial evidence that additional oncogenic viruses may have roles in breast cancer. These are the bovine leukaemia virus (BLV), the high risk human papilloma viruses (HPV) and the Epstein–Barr virus (EBV) [144]. Each of these viruses has been identified in normal and benign breast tissues up to 11 years before the development of the same virus positive breast cancer in the same women [41]. In a series of studies, the Al Moustafa group has demonstrated that HPV and EBV frequently co-exist, and they may collaborate in human breast cancer [145]. There is also evidence that EBV and human papilloma viruses may interact, leading to increased cell proliferation [146].

As discussed above, the role of these viruses is complex and may be indirect in the same way that the human immunodeficiency virus (HIV) has an indirect role in Kaposi’s sarcoma.

### 12.3. Prevention

For the prevention of MMTV human breast cancer, preliminary experiments using MMTV p14 proteins on mice have been encouraging [147]. Several successful immunisation strategies against MMTV mouse mammary tumours were developed decades ago [148]. Immunization in mice may be more effective than it would be in humans, where other risk factors may play a major role.

Retroviral vaccines have proved to be difficult to develop. So far, it has not been possible to develop a successful vaccine against the human immunodeficiency retrovirus (HIV). This is because of the repeated mutation of HIV. However, the use of RNA-based vaccines to help control the COVID-19 pandemic has led to the possibility of an MMTV vaccine.

## 13. Conclusions

When considered overall, the evidence is compelling that MMTV has a probable causal role in a subset of up to 40% of human breast cancers.

## Figures and Tables

**Figure 1 viruses-14-00721-f001:**
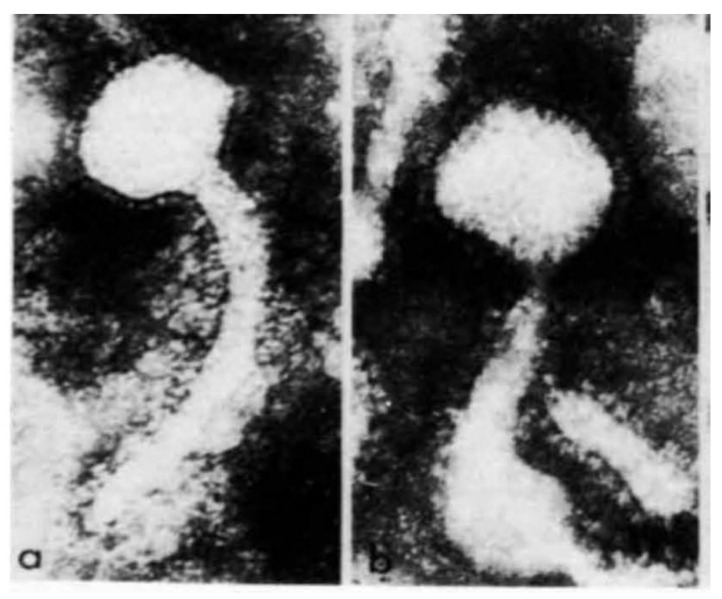
Particles found in human and mouse milk and in culture fluid. (**a**) Negatively stained particle from American human milk; (**b**) Negatively stained particle from RIll mouse milk (×18,000) [18]. (Republished with copyright permission from Nature).

**Table 1 viruses-14-00721-t001:** MMTV positive breast cancer compared to benign and normal breast controls.

Study	Country	Method	Breast Cancer	0.001 s	Significance for Difference, Breast Cancer versus Controls, *p* Value
Axel 1972 [19]	US	Molecular hybridisation	19/29 66%	0.001 s	0.004 s
Mesa-Tajada 1978 [35]	US	IHC	15/131 39%	0/18 0%	0.001 s
Wang 1995 [23]	US	PCR	121/314 38.5%	2/107 2%	
Etkind 2000 [26]	US	PCR	27/73 37%	0/35 0%	
Melana 2001 [36]	US	PCR	32/106 30%	1/106 1%	0.001 s
Melana 2002 [37]	Argentina	PCR	23/74 31%	1/10 10%	0.003 s
Ford 2003 [27]	Australia	PCR	19/45 42%	2/111 2%	0.001 s
Ford 2004 [38]	Australia	PCR	45/144 31%	0/111 0%	0.001 s
Zammarchi 2006 [39]	Italy	PCR	15/45 33%	0/8 0%	0.008 s
Hachana 2008 [40]	Tunisia	PCR	17/122 14%	0/122 0%	0.001 s
Lawson 2010 [41]	Australia	In situ PCR	33/74 45%	0/29 0%	0.001 s
Mazzanti 2011 [31]	Italy	PCR	47/69 68%	0/20 0%	0.001 s
Glenn 2012 [42]	Australia	PCR	39/50 78%	13/40 33%	0.045 s
Slaoui 2014 [43]	Morocco	PCR	24/42 57%	6/18 33%	0.312 ns
Cedro-Tanda 2014 [44]	Mexico	PCR	57/458 12%	72/458 16%	0.308 ns
Naushad 2014 [45]	Pakistan	PCR	83/250 29%	0/15 0%	0.001 s
Reza 2015 [46]	Iran	PCR	12/100 12%	0/100 0%	0.002 s
Shariatpanahi 2017 [47]	Iran	PCR	19/59 32%	3/59 5%	0.002 s
Al Dossary 2018 [48]	Saudi Arabia	PCR	6/101 6%	0/51 0%	0.082 ns
Seo 2019 [49]	Korea	PCR	12/128 9%	0/128 0%	0.013 s
Al Hamad 2020 [8]	Jordan	PCR	11/100 11%	0/20 0%	0.023 s
Periera 2020 [9]	Brazil	PCR	41/217 19%	30/196 15% Tissues adjacent to cancer	0.417 ns
Loutfy 2021 [10]	Egypt	PCR	38/50 76%	0/10 0%	0.001 s
Wang 2021 [11]	China	PCR	21/119 18%	2/50 4%	0.05 s
Khalid 2021 [12]	Pakistan	PCR	69/105 66%	2/15 13%	0.023 s
Gupta 2021 [13]	Croatia	PCR	5/70 7%	0/16 0%	0.056 ns

s—significant; ns—not significant

**Table 2 viruses-14-00721-t002:** Mouse mammary tumour virus breast cancer serology.

Study	Location	Method	Breast Cancer	Controls	Statistical Significance
Muller 1972 [85]	Germany	Immuno Fluorescence	75/228 33%	11/95 12%	0.002 s
Ogawa 1978 [86]	Japan	Immuno Fluorescence	26/43 60%	4/37 11%	0.001 s
Mehta 1978 [87]	India	Immuno Fluorescence	26/34 76%	0/10 0%	0.003 s
Witkin 1979 [88]	US	Virolytic Assay	11/65 17%	2/60 3%	0.001 s
Imai 1979 [89]	Japan	Immuno Fluorescence	49/89 55%	18/68 27%	0.020 s
Witkin 1980 [24]	US	Elisa	14/54 26%	5/63 8%	0.026 s
Day 1981 [25]	US	Elisa	27/145 19%	1/36 3%	0.026 s
Nagayoshi 1981 [90]	Japan	Hemaglutination	34/96 36%	3/59 5%	0.001 s
Tomana 1981 [91]	US	Immuno Fluorescence	56/137 41%	2/56 4%	0.001 s
Zotter 1981 [92]	Germany	Immuno Precipitation	84/367 23%	11/184 6%	0.001 s
Holder 1983 [93]	US	Viral Agglutination	41/52 79%	2/18 11%	0.004 s
Litvinov 1984 [94]	Russia	Radio Immune Assay	51/92 55%	3/94 3%	0.001 s
Chattopadhyah 1984 [95]	India	Hemaglutination	14/14 100%	0/13 0%	0.004 s
Kovarik 1989 [83]	Czech Slovakia	Immunoblotting	2/60 3%	0/60 0%	0.226 ns
Goerdert 2006 [2]	US	Immunoblotting	0/92		
Zhang 2020 [84]	Canada	Elisa GP 52	10/98 10%	2/98 2%	0.017 s

s—significant; ns—not significant

## Data Availability

Not applicable.

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
