# Peer review of "Mouse Mammary Tumour Virus (MMTV) in Human Breast Cancer—The Value of Bradford Hill Criteria"

_viruses, 2022, doi:10.3390/v14040721_

Round 1
Reviewer 1 Report
The paper has been improved. I do not have other comments.
Author Response
No Comments.
Reviewer 2 Report
Lawson and Glenn reviewed evidence on the involvement of MMTV in human breast cancer considering the Hill's criteria with extensions proposed by the authors.
Although the theme is of interest and a field of intense debate on the literature, the manuscript lack novelty and some assumptions seems very speculative. Also, although an association can be inferred, stablishing MMTV as a potential risk factor for breast cancer, the evidence for its role as a causal agent is still controversial and warrants a more careful discussion. Some points are highlighted bellow:
1) Some important references arguing against the involvement of MMTV in human breast cancer were not discussed, such as the work from Perzova et al. (10.1186/s12985-017-0862-x), which suggest that many of the reported MMTV detection in published studies may come from environmental contamination with murine DNA during PCR preparation. Although this study does not invalidate published results, it may suggest that the reported prevalence of MMTV is overestimated in some studies. It may also be an alternative explanation for the correlation between the MMTV prevalence in human breast cancer and the presence of Mus domesticus.
2) Also, regarding the discussion about possible contamination in some reviewed studies, must be more carefully discussed, since this contaminations can occur at many levels (e.g.: reagents, DNA, environmental contamination, PCR carryover), and many studies employ different methods to control for contamination beyond amplification of murine DNA. Also, the simple fact that some studies report a relatively high prevalence of MMTV DNA in control samples (that have different origin, such as tissue adjacent to tumor, or mammary gland from reduction mammoplasty) does not constitute a evidence for contamination, and this assumption is very speculative. Noteworthy, in another part of the text authors themselves highlight a prevalence of 11% in saliva of healthy adults and the presence of MMTV in benign lesions of women who develop breast cancer after in their lives. Therefore, it is not unexpected that some healthy women harbor MMTV in mammary tissue without having cancer. Even more acceptable is that tissue adjacent to tumors are infected by MMTV, since both are within the same host.
3) Authors should also discuss the recently published results from large global consortia on cancer genomics (ICGC and TCGA) that screened a large group of tumor samples from different origins and countries using NGS and found no evidence of MMTV infection in any analyzed tumor (Zapatka et al., 10.1038/s41588-019-0558-9).
4) The stretch between lines 271 and 275 seems excessively speculative.
5) When authors discuss the role of APOBEC3 family in the development of BC associated with MMTV, they highlight APOBEC3B as a main candidate. However, other members of APOBEC3 family may also have mutagenic role. Indeed it has been shown that APOBEC3A is a more potent mutagen than APOBEC3B, and that the increased APOBEC-mediated mutational rate observed in patients carrying a polymorphism completely deleting APOBEC3B is attributable to the stabilization of APOBEC3A expression (10.1038/ng.3378). Also none of these enzymes have been conclusively associated to MMTV infection in human BC. Therefore, the article may benefit from a deepened discussion on the role of APOBEC3 family in the carcinogenesis of MMTV-associated BC, and it must make it clear that this mechanism is still a unproved hypotheses raised by the authors and by previous studies on the field. Experimental data is still required to confirm this.
6) Although the involvement of other viruses in breast cancer is an interesting point to discuss since it may suggest shared mechanisms, I don't think it holds strength as a criterium to infer causality. It would be better fitted in the introduction or discussion of the paper as an additional evidence.
7) In the section "Prevention" (lines 531-540) authors must make clear that the discussed data refer to murine models. The way it is written it gives the impression that the promising results were obtained in human volunteers, specially when reading the first phrase of the section: "For the prevention of MMTV human breast cancer, preliminary experiments using 532 MMTV p14 proteins have been encouraging [103]". Also note that MMTV is the causative agent of mammary tumors in mice, therefore immunization in these animals may be more effective in preventing breast cancer than it would be in human, where other risk factors play major roles.
Author Response
Please see attachments (i) responses to the Academic Reviewer plus detailed responses to the Second Reviewer and (ii) revised manuscript using the existing MDPI formatted manuscript.
Please note the MDPI format uses right sided "justification". This cuts some word in half and alters their meaning. This has not been corrected.

Round 2
Reviewer 2 Report
The manuscript have improved with modifications provided by the authors.
Additional points to consider and adjust:
1) Overall the manuscript has great similarity with many previous review articles reviewing the evidence for MMTV causal role in human breast cancer based on Hill's criteria, including works from the authors themselves (DOIs: 10.1186/s42269-020-00439-0; 10.1038/s41523-019-0136-4; 10.1016/j.micpath.2019.03.021; 10.1186/s13027-021-00366-3; 10.1016/j.canlet.2018.01.076). It is important to reference these previous works and make clear if the present work is an update on previous discussions based on new reports on the literature in the introduction section and to discuss similar and contradictory conclusions among the current work and these previous reviews in the discussion section if that is the case.
2) Use of italic for scientific names (such as Mus musculus and Musa domesticus)
3) Line: 446 "Elisa gp52" should be corrected to "ELISA to gp52".
4) Line 627: SARS-CoV-2, the etiological agent of COVID-19, is a +ssRNA virus, and not a retrovirus, as stated.
Author Response
Please find attached response

This manuscript is a resubmission of an earlier submission. The following is a list of the peer review reports and author responses from that submission.
Round 1
Reviewer 1 Report
In my opinion this review article is very similar to what already exists in the literature, so I don't understand the need for a publication with everything that has already been written; it could just be a mini review where the authors could cite what is new on this topic .
Reviewer 2 Report
The paper analyzes the data available on MMTV and breast cancer having Hill’s criteria as a model. This should be explained from the beginning, maybe also in the title. “Mouse mammary tumor virus (MMTV) in human breast cancer: the going relevance of Hill’s criteria” or similar.
Not many people know Hill’s criteria. Authors could remind the original 9 ones and the fact that they were introduced to give strength to the studies on occupational medicine. They say that Vonka added a series of tests but they do not explain which ones and do not mention the two lists of criteria cited in the paper, proposed by Evans and by zur Hausen. The list of 12 criteria they propose/adopt is not explained. The criteria should be general and then adapted to MMTV. For instance, the first one “identification and history of MMTV in human breast cancer” is not a general concept and does not give strict rules. The presence of MMTV by itself is not relevant (could be secondary to contamination) and what “history of MMTV in human breast cancer” means is not clear at all. A different definition would be better.
By the way, Authors could explain that Austin is the first name, Bradford the middle name and Hill the last name. In some papers he appears as Bradford-Hill.
The paper is very interesting. However, it is so plenty of information that makes it difficult to read. Authors should try to eliminate information not strictly relevant. For instance, the subject of epidemiology requires to be reconsidered. An important information is that all events that make the period of fertility longer increase the risk of cancer because they increase the time of exposure to estrogens. Estrogens are the most powerful factor of promotion for breast cancer. Similarly, obesity increases the endogenous production of estrogens. But this has no relationship with breast cancer etiology, with MMTV in particular.
For what concerns cancer incidence and migrations, old data are not reliable. Data about variation in risk in populations who move to a country with a different incidence do not have strong biological basis and recently are questioned:
- “These differences likely reflect differences in cancer risk factors, in the use of screening tests, and in other preventive and treatment interventions”. Jeremiah Hwee, and Evelyne Bougie 2021 Do cancer incidence and mortality rates differ among ethnicities in Canada? DOI: https://www.doi.org/10.25318/82-003-x202100800001-eng
- Lower attendance rate in immigrants in screening programs. Bhargava S et al 2018 J Med Screen 2018, Vol. 25(3) 155–161
- Gomez SL et al 2010. These findings challenge the notion that breast cancer rates are uniformly low across Asians and therefore suggest a need for increased awareness, targeted cancer control, and research to better understand underlying factors. (Am J Public Health. 2010;100:S125–S131. doi:10.2105/AJPH.2009.163931)
- Morey BN et al Prev Chronic Dis 2019;16:180221. DOI:https//doi.org/10.5888/pcd16.180221 Higher Breast Cancer Risk Among Immigrant Asian American Women Than Among US Born Asian American Women. “However this finding is contrary to earlier studies of Asian American populations in California showing hat immigrants had lower rates of breast cancer than US born women”.
Honestly, I do not see any relation amongst these epidemiological data, Hill, MMTV.
I suggest to edit the entire paper taking out information not strictly related to the topic and using a more concise style. Figs 2 and 3 are not very explicative, maybe can be eliminated.
In lines 42-45 the word “evidence” is repeated 5 times.
In summary: the text should to be a little shorter and more fluid and data not really relevant should be taken out.